# Outdoor Time, Space, and Restrictions Imposed on Children’s Play in Australian Early Childhood Education and Care Settings during the COVID Pandemic: A Cross-Sectional Survey from Educators’ Perspective

**DOI:** 10.3390/ijerph20186779

**Published:** 2023-09-18

**Authors:** Junjie Liu, Shirley Wyver, Muhammad Chutiyami, Helen Little

**Affiliations:** 1Macquarie School of Education, Macquarie University, Sydney, NSW 2109, Australia; shirley.wyver@mq.edu.au (S.W.); muhammad.chutiyami@uts.edu.au (M.C.); helen.little@mq.edu.au (H.L.); 2School of Nursing and Midwifery, Faculty of Health, University of Technology Sydney, Sydney, NSW 2007, Australia

**Keywords:** outdoor activity, outdoor time, outdoor space, COVID-19, Australian early childhood education and care settings, educators’ tolerance of risk

## Abstract

The study aimed to investigate the impact of COVID-19 restrictions on children’s access to the outdoors in early childhood education and care (ECEC) settings. An online survey comprised of a standardised scale and questions used in previous related studies was completed by 143 early childhood educators across Australia. Participants were asked to report children’s time spent outdoors, access to outdoor space, and restrictions imposed on children’s play during the pandemic compared to the pre-pandemic period. The survey responses were imported into SPSS for descriptive, correlation, and ordinal regression analyses. We examined the relationship between children’s outdoor time, space, restrictions imposed on children’s play, and educators’ characteristics, such as qualifications, professional development, and tolerance of risk or staff–child ratios during the pandemic in Australia. Most educators reported that children’s outdoor time and access to outdoor space remained the same compared to before the pandemic, while imposed restrictions on children’s outdoor play increased. The results of ordinal regressions indicated that educators with a higher tolerance of risk were less likely to impose additional restrictions on children’s outdoor play. The findings contribute to the understanding of how educators’ tolerance of risk influences children’s outdoor play opportunities and provide directions for future risk-reframing interventions.

## 1. Introduction

On 11 March 2020, COVID-19 was declared a pandemic [1]. Governments worldwide advocated a range of behavioural interventions and restrictions to reduce infection rates. The closure of schools and limited access to parks, playgrounds, natural areas, and recreational facilities restricted the possibility for children to be outdoors [2]. COVID-19, as a sudden emergence, affected all families, schools, and children [3]. There were concerns regarding children’s restricted opportunities for outdoor play [4]. Australia detected its first COVID-19 case on 25 January 2020, after many countries had developed safety protocols. In 2020, similar restrictions across states/territories were put in place to reduce the spread of the pandemic, including restricting large gatherings and keeping social distance [5,6]. Policies in response to COVID-19 varied among states and territories during 2021, with New South Wales and Victoria implementing stringent measures [6]. Early childhood education and care (ECEC) settings refer to day-care centres, preschool, and childcare in Australia. During the pandemic, ECEC services in Australia were required to stay open [7]. Australian state and territory governments launched new guidelines in areas such as hand hygiene, social distancing, minimising indoor time, and mask-wearing, to reduce COVID-19 transmission [8,9,10]. While the restrictions were intended to control the spread of the virus, the impact on quality outdoor play experiences for children in ECEC remains uncertain [11].

Numerous studies have investigated the impacts COVID-19 had on children’s outdoor activity. A recent systematic review [12] showed that COVID-19 restrictions were associated with outdoor activity in ECEC contexts internationally, but that outdoor activity was underrepresented in the research. The two studies [13,14] with a focus on ECEC settings identified an increase in outdoor activity and that children wanted to increase their outdoor time. These studies had a focus on the amount of time children spent outdoors. Other potential changes that might result from COVID-19 restrictions were not included. For example, although children may be spending more time outdoors, there may be restrictions on access to particular play areas (e.g., sectioning off areas with equipment that does not meet new hygiene requirements) and restrictions on children’s behaviours to encourage social distancing. Additionally, there are no published investigations of the characteristics of educators that might predict specific changes in outdoor environments due to COVID-19. As we will discuss, educators’ tolerance of risk, qualifications, and professional development in outdoor play may be influential in determining changes in outdoor environments in response to new COVID-19 regulations. Factors such as staff–child ratios are also likely to be important.

Risky outdoor play has been found to be associated with children’s overall healthy development [15]. There is emerging evidence that tolerance of children’s risk-taking may influence how educators perceive and value outdoor play. Tolerance of risk refers to the degree to which risk in play is accepted by adults. Hill and Bundy [16] introduced a tool for measuring risky play tolerance to gain a better understanding of variations in tolerance of risk. The Tolerance of Risk in Play Scale (TRiPS) reflects Sandseter’s [17] six categories of risky play and yields valid and reliable data for measuring adult tolerances of risk during children’s play. Ihrig [18] revised the TRiPS to Teacher Tolerance of Risk in Play Scale (T-TRiPS) for use with prior-to-school teachers. The T-TRiPS can serve as a valuable tool not only for teachers to gain insight into their own risk tolerance but also as a basis for interventions aimed at shifting teachers’ perspectives and increasing risk tolerance. The present study included the T-TRiPS to examine whether there is a relationship between educators’ tolerance of risk-taking and children’s outdoor activity.

There are mixed findings regarding qualifications and educator outdoor play practices. Qualifications were associated with Canadian educators meeting outdoor play best practices [19], yet Hwang et al. [20] found that educators’ qualifications did not contribute to outdoor play changes within Canadian childcare centres during the pandemic. The inconclusive findings indicated that further research on the relationship between educator qualifications and children’s outdoor play in ECEC settings is needed. In Australia, there are three types of early childhood qualifications: certificate III level, diploma level, and early childhood teaching level [21]. Some studies conducted in Australia have obtained information on educators’ qualifications but have not included it as a factor to examine the influences on the quality of interactions in outdoor environments or the amount of outdoor time educators allowed children to access outdoor play areas [22,23]. It is therefore important to consider if there is a relationship between educators’ qualifications and children’s outdoor activity in ECEC settings.

The findings of a Canadian study indicated that more frequent professional development was associated with meeting best outdoor play practices [19]. Mohr [24] found a positive relationship between US teachers’ professional development and children’s outdoor risky play experience. Australian educators indicated that professional development could improve their skills to facilitate outdoor physical activities [25]. However, there is a lack of quantitative evidence regarding the relationship between professional development and children’s outdoor play within Australian ECEC contexts. Research conducted in Canada examined the relationship between physical activity (PA) and professional development. The results showed that PA training for early childhood educators could positively influence their self-efficacy to engage preschoolers in PA and increase their knowledge of PA guidelines, which led to an improvement in both the quantity and quality of active play opportunities offered to young children in childcare [26]. As physical activity is usually associated with children’s outdoor play [27], there is some evidence that professional development predicts outdoor play opportunities [19].

Staff–child ratios were also found to be positively associated with process quality, indicating that better interactions with educators, peers, materials, or activities were observed when there were fewer children per staff member [28]. Australian educators perceived that having fewer children per staff member provided extra outdoor play opportunities for children [25]. Educators in the UK also indicated that adequate staff–child ratios might lead to increased interactions with children outdoors or utilisation of the outdoor environment [29]. It is important to note that limited studies have examined the relationship between staff–child ratios and outdoor play opportunities, and the existing studies were conducted using qualitative methods. Therefore, further investigation is needed to understand how staff–child ratios may impact outdoor play in ECEC settings.

The present study examined the impact of COVID-19 restrictions on children’s opportunities to engage in outdoor activity. We addressed the following questions:

Compared to pre-pandemic conditions:
Was there a change in the amount of time children spent outdoors?Was there a change in the ECEC space provided for outdoor activity?Did educators change their restrictions on outdoor play?Were any changes in time, space, and restrictions related to educator characteristics or quality indicators (staff qualifications, professional development, tolerance of risk, and staff–child ratios)?

## 2. Materials and Methods

The current study includes the analysis of survey data from a larger PhD study by the first author on the impact of COVID-19 restrictions on outdoor activity in Australian ECEC settings. Ethics approval was obtained from Macquarie University (Reference No: 520221188140234).

### 2.1. Participants

The study was conducted through an online survey distributed to ECEC services across Australia via an online professional magazine (Early Childhood Australia Web Watch), Facebook, and email. The survey was posted in the magazine and on Facebook groups with an active LimeSurvey link. Emails were sent to centres identified by the StartingBlocks website, the government database for ECEC services in Australia. Educators working from the pre-pandemic period onwards in Australian ECEC were eligible to participate. The survey was available from 17 October to 31 December 2022. Participants were educators working from the pre-pandemic period onwards in Australian ECEC services across various regions, including New South Wales (30.1%), Victoria (35.0%), Queensland (16.1%), Western Australia (4.9%), South Australia (5.6%), and the Australian Capital Territory (8.4%). A total of 146 educators participated in the online survey through the LimeSurvey platform. The final sample for analysis comprised 143 responses. Three responses were not eligible and were deleted as the children were aged 6–12 years or the educators did not complete all the questions.

### 2.2. Survey

The online survey included 50 questions designed to collect potential changes in children’s outdoor time, space, and restrictions imposed on children’s play during the pandemic in Australian ECEC and to measure educators’ tolerance of risk. The survey questions were developed by drawing upon a systematic review [12] and a validated teacher tolerance of risk scale [18]. The scale was modified for the purpose of the study. There are also six questions from other relevant publications [19,30,31,32]. The online survey was divided into three parts: educator background and centre characteristics, potential changes in outdoor activity, and educators’ tolerance of risk. All the authors checked and pre-tested the online survey in English to guarantee flow and clarity.

#### 2.2.1. Educator Background and Centre Characteristics

The first section included 13 questions related to educator background and centre contexts, such as gender, age, working experience, qualifications, professional development, educators’ knowledge of COVID-19, centre location, state, age groups, staff–child ratios, and outdoor space. The questions were based on previous research conducted with early childhood educators [19,30,31,32]. The exceptions were the two questions regarding educators’ knowledge of COVID-19, which were developed by the authors to determine if educators were aware of the health advice that being outdoors lowered the risk of infection and to determine where they had obtained this information.

#### 2.2.2. Potential Changes in Outdoor Activity

Eight questions were used to explore potential changes in children’s outdoor activity. The questions had previously been used in published studies examining children’s outdoor activity changes during the pandemic [33,34]. Educators were asked to compare children’s outdoor time and access to outdoor space before and during the pandemic. We used a five-point Likert scale to assess outdoor time ranging from ‘a lot less’ to ‘stay the same’ to ‘a lot more’. The educators were asked to report if there were any changes in children’s outdoor time compared to before the pandemic. There was a yes/no response regarding potential changes in children’s access to outdoor space. Those who reported changes were asked to specify if they were decreases, increases, or anything else.

#### 2.2.3. Educators’ Tolerance of Risk

The T-TRiPS was used to measure educators’ tolerance of risk, with 25 questions requiring a yes/no response and providing an option for additional comments [18]. The T-TRiPS was developed in the USA and not all items were suitable culturally or in terms of Australian ECEC regulations. Five questions were modified to align with Australian ECEC contexts, five were removed, and two were added, resulting in a total of 22 questions. An example of a modification is the change from ‘do you allow your students to continue playing after s/he gets a scrape?’ to ‘do you allow children to continue playing if they fall and get up?’. Participants who indicated working with the birth–three-year-old group were not required to finish this part as the T-TRiPS is designed for those working with pre-schoolers. The adapted scale indicated a good internal consistency (N = 110, Cronbach α = 0.84), suggesting an acceptable reliability of the scale. Furthermore, three questions structured similarly to those of the T-TRiPS, relating to social distancing, physical contact, and the number of children playing together, were included. These questions were developed to examine if educators imposed restrictions on children’s risky/boisterous play during the pandemic. The three questions also showed a good internal consistency (N = 110, Cronbach α = 0.74), further indicating the extent to which the three questions were reliable in measuring the restrictions educators imposed on children’s play.

### 2.3. Data Analysis

The survey data were exported from LimeSurvey to Excel for the purpose of data cleaning and coding. The numerical data were imported into IBM SPSS Statistics version 28 for descriptive, correlation, and inferential analysis. Spearman correlations were conducted to examine the individual relationships between the study variables. We conducted ordinal regression to examine the relationship between each of the three dependent variables and four independent variables. The dependent variables include outdoor time, measured by a five-point scale, access to outdoor space, and restrictions imposed on children’s play during the pandemic. The independent variables include educators’ qualifications, professional development, staff–child ratios, and T-TRiPS. Two authors (JL, SW) coded responses to an open question about professional development to determine if it was a one or two. For the ordinal regression, we applied a complementary log–log link function for time spent outdoors and access to outdoor space, as inspection of the distribution of the responses showed a prevalence of frequencies in higher categories. In contrast, the frequencies focused more on lower categories for restrictions imposed on children’s play, and we applied the negative log–log link function.

Educators reported intersecting age groups with irregular categories from under three to over three to mixed ages. As a result, we were unable to use age as a variable in our analyses. However, age could be an important predictor. So, we conducted the Kruskal–Wallis H test to examine if age groups were significant predictors of time spent outdoors, access to outdoor space, and restrictions imposed on children’s play during the pandemic. The results indicated no significant differences for age group and changes in (1) time spent outdoors (χ^2^(2, N = 140) = 5.14, *p* > 0.05); (2) children’s access to outdoor space (χ^2^(2, N = 140) = 0.29, *p* > 0.05); or (3) restrictions imposed on outdoor play (χ^2^(2, N = 109) = 1.69, *p* > 0.05).

## 3. Results

### 3.1. Educator Background and Centre Context

Table 1 provides a summary of educators’ backgrounds and the centre context where they work. There were 143 responses in total, of which 132 were completed surveys and 110 included valid responses regarding educators’ tolerance of risk. Eleven participants skipped the last part of the survey regarding how difficult it was to recall the outdoor practice during the pandemic. Of those who completed the 5-point Likert scale, recalling outdoor practice during the pandemic was rated as (1) not difficult (56.1%); (2) a little bit difficult (20.5%); (3) rather difficult (13.6%); (4) very difficult (7.6%); (5) extremely difficult (2.3%). Most participants did not find it difficult to recall outdoor play practice during the pandemic.

### 3.2. Outdoor Time, Outdoor Space Changes, and Restrictions during the Pandemic in Australian ECEC

As shown in Table 1, there were changes in time spent outdoors reported by educators as an increase or a decrease. However, most educators indicated that children’s outdoor time stayed the same during the pandemic. Similarly, when it came to children’s access to outdoor space, a larger number of educators reported no changes. It is also apparent from Table 1 that a greater number of educators imposed restrictions on children’s play in response to COVID-19.

The Spearman’s Rank Order correlations revealed that none of the dependent variables were highly correlated, as can be seen in Table 2. Therefore, all were acceptable to include in our regression analyses. Professional development and staff–child ratios were positively correlated with outdoor time. Professional development and educators’ tolerance of risk were negatively associated with imposing restrictions on children’s play.

### 3.3. Relationships between Changes in Outdoor Time, Space, Restriction, and Individual Characteristics or Quality Indicators

Three regression analyses were conducted to predict changes in time spent outdoors, access to outdoor space, and restrictions imposed on children’s play. We used ordinal regression as outdoor time, space, and restrictions were all ordinal variables. We conducted the analyses with educators’ qualifications, professional development, staff–child ratios, and tolerance of risk as the independent variables. The following tables present the results of the regression analyses for changes in outdoor time, space, and restrictions imposed on children’s play.

As shown in Table 3, there were no significant results regarding children’s time spent outdoors, educators’ individual characteristics, or staff–child ratios.

There were no significant predictors of children’s access to outdoor space found in the results presented in Table 4.

Table 5 showed that educators’ tolerance of risk was the only significant factor in the model. Educators with a higher tolerance of risk were less likely to impose restrictions on children’s outdoor play during the pandemic. Educators’ qualifications, professional development, and staff–child ratios were not significant variables in predicting the extent to which educators restricted children’s play due to the pandemic. The model explained 19% of the variation in the dependent variable.

## 4. Discussion

This study examined if there were any changes in outdoor time, space, and restrictions imposed on children’s play in Australian ECEC during the pandemic. Almost all the educators were aware of the health advice that being outdoors could minimise the risk of virus transmission. However, the findings indicated that more educators kept the same amount of outdoor time during the pandemic and did not change children’s access to outdoor space. We also found that most educators imposed restrictions on children’s outdoor play compared to before the pandemic. The findings from the current study were inconsistent with two studies included in a systematic review [12,13,14] that examined the time children spent outdoors in ECEC settings, which reported increased outdoor time during the pandemic. The overall results demonstrated that children’s time spent outdoors and access to outdoor space remained the same, as reported by educators. However, educators reported changing their practices by implementing restrictions on children’s play to reduce physical contact. As a result, the quality of children’s outdoor time might have been impacted when they spent the same time outdoors and had access to outdoor space but while their behaviours were restricted, such as their movements or rough-and-tumble play. Therefore, further research investigating the reasons behind the changes/no changes in children’s outdoor activity is needed to support the findings regarding educators’ practices in ECEC services.

Changes in restrictions imposed on children’s play were found to be associated with educators’ tolerance of risk. There were no significant ordinal regression results regarding educators’ qualifications, professional development, and staff–child ratios. Educators with a higher tolerance of risk were less likely to impose restrictions on children’s outdoor play during the pandemic. Therefore, more attention should be paid to risk reframing interventions. Recent research conducted with risk reframing tools indicated that the interventions were efficient in increasing early childhood educators’ tolerance of risk or supporting their outdoor play pedagogy [35,36]. Even though educators in Australia value children’s opportunities to engage in risky play [37], the findings of the study suggested that risk reframing interventions may serve as an important measure in supporting educators to reduce the restrictions imposed on children’s play and provide more outdoor play opportunities for children.

To the best of our knowledge, no study has been conducted to examine children’s outdoor time, access to outdoor space, and restrictions imposed on children’s play during the pandemic and their influential factors. Therefore, the results of the study were exploratory and preliminary. Even though educators’ tolerance of risk was a significant predictor of restrictions imposed on children’s outdoor play during the pandemic. It is important to note that there are other variables that have not been considered in the model. The findings of the study support the importance of educators’ tolerance of risk, but it has not provided the overall picture.

The current study adds evidence to the existing literature by showing the importance of educators’ tolerance of risk in creating more opportunities to play outdoors and the difficulties in changing outdoor play practices. The findings provide directions for future risk reframing interventions. However, the current study had limitations. First, it was a cross-sectional study, as the data were collected after the COVID-19 restrictions and at a relatively late stage of the pandemic in Australia and relied on a self-reported survey. Despite the majority of educators reporting no difficulty in recalling outdoor play practices, there were possibilities of recall biases. Second, the limited sample size restricted the number of independent variables to be included in the model. Third, the survey responses were not collected from all states/territories in Australia. The sample cannot be considered representative. Most participants were located in the eastern part of the country, where COVID-19 restrictions were more stringent. Even with these limitations, the current study contributes to the understanding of how educators’ tolerance of risk influences restrictions imposed on children’s play during the pandemic.

## 5. Conclusions

COVID-19 restrictions abruptly impacted children’s daily life and their access to the outdoors, but there is little evidence about what occurred within the ECEC context. The present study examined the impact of COVID-19 restrictions on the time children spent outdoors, their access to outdoor space, and the restrictions imposed on children’s play in Australian ECEC and their influential factors. The results showed that most educators maintained the same amount of time spent outdoors and access to outdoor space despite awareness of the health advice that being outdoors lowered the risk of infection. There were restrictions imposed on children’s outdoor play due to the pandemic, which might have impacted the quality of children’s outdoor time. Educators with a higher tolerance of risk tended to impose fewer restrictions on children’s play. Therefore, we believe that risk reframing interventions may play an important role in decreasing the restrictions imposed on children’s outdoor play and providing more opportunities for children to play outside in ECEC services.

## Figures and Tables

**Table 1 ijerph-20-06779-t001:** Educator background and centre context.

Background	Response	Number/Percentage
Gender	Female	134/93.7%
	Male	7/4.9%
	Other (non-binary educator)	2/1.4%
	Total	143
ECEC working experience	1–3 years	19/13.3%
	4–7 years	25/17.5%
	>7 years	99/69.2%
	Total	143
Qualifications	Non-degree	80/55.9%
	Degree	59/41.3%
	Total	139
Professional development on outdoor play	No professional development related to outdoor play	85/59.4%
	Received professional development in outdoor play as part of getting the qualifications or did not specify	34/23.8%
	Received professional development related to outdoor play	24/16.8%
	Total	143
Aware of health advice on staying outdoors	Yes	137/95.8%
	No	6/4.2%
	Total	143
Age group	Under 3 years old	33/23.1%
	Over 3 years old	40/28.0%
	Mixed age	67/46.9%
	Total	140
State/territory	New South Wales (NSW)	43/30.1%
	Victoria (VIC)	50/35.0%
	Queensland (QLD)	23/16.1%
	Western Australia (WA)	7/4.9%
	South Australia (SA)	8/5.6%
	Australian Capital Territory (ACT)	12/8.4%
	Total	143
Outdoor space	Fewer than 7 square metres per child	18/12.6%
	7 square metres per child	59/41.3%
	More than 7 square metres per child	57/39.9%
	Other (not sure, or extremely large)	9/6.3%
	Total	143
Staff–child ratio	1:3	3/2.1%
	1:4	48/33.6%
	1:5	20/14.0%
	1:6	2/1.4%
	1:7	4/2.8%
	1:8	4/2.8%
	1:9	1/0.7%
	1:10	22/15.4%
	1:11	18/12.6%
	1:15	1/0.7%
	Total	123
Outdoor time	A lot less	6/4.2%
	A little less	16/11.2%
	About the same	64/44.8%
	A little more	35/24.5%
	A lot more	22/15.4%
	Total	143
Outdoor space changes	Decrease	5/3.5%
	Stay the same	87/60.8%
	Increase	51/35.7%
	Total	143
Educators’ tolerance of risk	T-TRiPS total	110
Restrictions imposed on children’s outdoor play	No restrictions	46/41.8%
	A few restrictions	24/21.8%
	Medium restrictions	17/15.5%
	More restrictions	23/20.9%
	Total	110

**Table 2 ijerph-20-06779-t002:** Spearman’s Rank Order correlations of changes in outdoor time, space, restrictions, and individual characteristics or quality indicators.

Study Variables	(1)	(2)	(3)	(4)	(5)	(6)	(7)
(1) Outdoor time	-						
(2) Outdoor space	0.153	-					
(3) Restrictions imposed on children’s outdoor play	0.194 *	0.105	-				
(4) Qualifications	0.137	−0.142	−0.078	-			
(5) Professional development	0.174 *	−0.115	−0.221 *	0.220 **	-		
(6) Staff–child ratio	0.243 **	−0.058	0.004	0.375 **	0.171	-	
(7) Educators’ tolerance of risk	0.039	−0.158	−0.298 **	0.130	0.324 **	0.007	-

Note: * *p* < 0.05, ** *p* < 0.01.

**Table 3 ijerph-20-06779-t003:** Ordinal regression analysis for changes in outdoor time (N = 110).

Independent Variables	Estimate	Std. Error	*p*-Value	95% Confidence Interval
Qualifications	−0.11	0.26	0.66	−0.63–0.40
No professional development	−0.58	0.38	0.12	−1.32–0.16
Professional development as part of qualification/not specified	–0.65	0.40	0.10	−1.43–0.12
Professional development related to outdoor play	* reference	* reference	* reference	* reference
Staff–child ratio	0.05	0.04	0.22	−0.03–0.13
Educators’ tolerance of risk	−0.03	0.03	0.26	−0.09–0.03

Pseudo R-square: Nagelkerke 0.08 * reference category group.

**Table 4 ijerph-20-06779-t004:** Ordinal regression analysis for changes in outdoor space (N = 110).

Independent Variables	Estimate	Std. Error	*p*-Value	95% Confidence Interval
Qualifications	0.02	0.31	0.96	−0.59–0.62
No professional development	0.15	0.40	0.71	−0.64–0.94
Professional development as part of qualification/not specified	−0.00	0.42	1.00	−0.83–0.83
Professional development related to outdoor play	* reference	* reference	* reference	* reference
Staff–child ratio	−0.03	0.05	0.56	−0.12–0.07
Educators’ tolerance of risk	−0.06	0.04	0.11	−0.13–0.01

Pseudo R-square: Nagelkerke 0.05 * reference category group.

**Table 5 ijerph-20-06779-t005:** Ordinal regression analysis for changes in restrictions imposed on children’s play (N = 110).

Independent Variables	Estimate	Std. Error	*p*-Value	95% Confidence Interval
Qualifications	0.27	0.30	0.38	−0.33–0.86
No professional development	0.56	0.48	0.25	−0.38–1.49
Professional development as part of qualification/not specified	0.02	0.53	0.97	−1.01–1.05
Professional development related to outdoor play	* reference	* reference	* reference	* reference
Staff–child ratio	0.01	0.05	0.89	−0.09–0.10
Educators’ tolerance of risk	−0.09	0.03	0.008	−0.16–−0.02

Pseudo R-square: Nagelkerke 0.19 * reference category group.

## Data Availability

The data presented in this study are available on request from Junjie Liu (junjie.liu2@students.mq.edu.au).

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
