# Peer review of "Outdoor Time, Space, and Restrictions Imposed on Children’s Play in Australian Early Childhood Education and Care Settings during the COVID Pandemic: A Cross-Sectional Survey from Educators’ Perspective"

_ijerph, 2023, doi:10.3390/ijerph20186779_

Round 1
Reviewer 1 Report
The exploration of the theoretical background is coherent and informative, but at the same time concise. The references are relevant, the bibliographic data are adequate.
The authors justify the relevance of the topic of the study in several places by pointing out the unique and gap-filling nature of their investigation.
Row 82-84: based on the list, it is not clear to which qualification level the Australian training types belong, I would recommend that the authors provide the ISCED classification of each level.
The research questions are specific and well-defined, in line with the objectives and measuring instrument of the study.
I feel that the number of the sample is a bit low, and the sampling cannot be considered representative. At the same time, it seems from the text of the study that the authors made a serious effort to get the online questionnaire to as many interested parties as possible, and it was possible to fill it out for a relatively long time (almost 2.5 months).
The presentation of the used measuring tool is easy to follow, and the description of the modifications is sufficiently detailed. The statistical methods used are appropriate.
The presentation of the results is clear, the tables are informative, although it should be indicated in Table 2 that the values are (presumably) the values of r (correlation coefficient).
In the discussion, the authors deal with the generalizability of the results and their limitations, and also draw attention to the further investigation possibilities of the "risk of tolerance".
Overall, the study is a high-quality work that meets the scientific criteria and, by virtue of its exploratory nature, directs attention to a new area of research.
Reviewer 2 Report
The topic of whether there is emerging evidence that tolerance of children’s risk-taking may influence how educators perceive and value outdoor play seems to be an important one specifically during Covid-19. Yet, we have no proof that it still might be important.
Reviewer 3 Report
Thank you for the opportunity to read your article. The paper is well written, shows a good discussion with the theory and the methodology is clear presented.
I have some small remarks:
- First sentence of Abstract seems odd. I like that you present the aim (if it is from the paper), but is it really "your" aim (because you write "our") or is is the aim of the paper? There should be a connection to the second sentence to make it clear that the survey was conducted to reach this aim (I assume).
- Different writings of Covid/COVID-19 - please use the same.
- "The adapted scale indicated a good internal consistency (N= 110, Cronbach α = 0.84)." - could you provide some more details about this, also about the other three.
- "Furthermore, three questions structured similarly to T-TRiPS, relating to social distancing, physical contact, and the number of children playing together." - I don't understand the sentence, is there a word missing?
- "Table 1 also showed that a greater number of educators imposed restrictions on children’s play in response to COVID-19." - I would rather rewrite that sentence (and the similar ones following), it sounds like the table shows this and not the research. Maybe "Regarding ... it can be seen in Table 1 that ..."
- At the description of the correlations I was wondering why some of the correlations were mentioned in the text, but not all. Also please add an explanation for the stars and add in the table caption that you used the Spearman’s Rank Order.
- Table 3, 4, 5: Add to the Table caption "changes" like "... for changes in outdoor time ..." to clarify. What does the "reference" mean?
Reviewer 4 Report
The article Outdoor Time, Space, and Restrictions Imposed on Children’s Play in Australian Early Childhood Education and Care Settings during the Covid Pandemic presents a well-structured study with good conclusions, the main one being that "Educators with higher tolerance of risk tended to impose fewer restrictions on children's play" (l. 325-326, p.10).
There are, however, a few apparent issues to consider:
- The title does not clearly express what the study is about - there is no evaluation of the outdoor time and space made available to children in ECEC, and the data is obtained from the perception of educators
- Outdoor activity seems to be an unclear keyword;
- the characteristics of educators, namely educators' tolerance of risk, should be made explicit in the title;
- It would be necessary for the objectives to be formulated coherently throughout the paper: l.12-13, l114-122, p.318-320. Each of the formulations suggests nuances in what has been studied.
- Out of curiosity, hasn't Australia been in lockdown periods in which the ECEC was closed?
- Revise the statement on l.102-103; it seems to be an abusive inference, or refer to studies that support it;
- lack of clarification in some methodological procedures: how many educators were contacted, or possible to recruit for the study (l.133-134)?; the construction and validation of the survey are not at all precise; it is understood that in the T-TRiPS the scale of this survey was modified for the study(?); the changes made (l.176-177) were not validated by pre-test and factor analysis?
- In the results, the presentation of tables 3 and 4 does not seem relevant; the statement l.229-231, referring to a 'greater number of educators' which appears to be 40 out of 110;
- In the discussion section, the authors should bear in mind that the statements regarding the data obtained are the result of educators' perceptions, so statements such as "The overall results demonstrated that children's time spent outdoors and access to outdoor space remained the same" (l.274-275) do not seem appropriate; nor does it seem that the information gathered can support the statement on l.276-279. The same situation in the conclusions - l.323-324.
Round 2
Reviewer 4 Report
I have read the revised manuscript and the letter from the authors.
I thank the authors for responding to my earlier comments and for making the revisions described. The authors have revised the paper adequately (including the title) and improved it. I have no additional comments on the article.
I will leave the decision about publication to the editors